# Remotely Sensed Mid-Channel Bar Dynamics in Downstream of the Three Gorges Dam, China

**Zhaofei Wen** [1,2,*] **, Hong Yang** [3,4] **, Ce Zhang** [5] **, Guofan Shao** [2] **and Shengjun Wu** [1]

1    Key Laboratory of Reservoir Aquatic Environment, Chongqing Institute of Green and Intelligent Technology, Chinese Academy of Sciences, Chongqing 400714, China; wsj@cigit.ac.cn
2    Department of Forestry and Natural Resources, Purdue University, West Lafayette, IN 47906, USA; shao@purdue.edu
3    State Key Laboratory of Lake Science and Environment, Nanjing Institute of Geography and Limnology, Chinese Academy of Sciences, 73 East Beijing Road, Nanjing 210008, China; h.yang4@reading.ac.uk
4    Department of Geography and Environmental Science, University of Reading, Whiteknights, Reading RG6 6AB, UK
5    Lancaster Environment Centre, Lancaster University, Lancaster LA1 4YQ, UK; c.zhang9@lancaster.ac.uk
*    Correspondence: wenzhaofei@cigit.ac.cn

**Abstract:** The downstream reach of the Three Gorges Dam (TGD) along the Yangtze River (1560 km) hosts numerous mid-channel bars (MCBs). MCBs dynamics are crucial to the river's hydrological processes and local ecological function. However, a systematic understanding of such dynamics and their linkage to TGD remains largely unknown. Using Landsat-image-extracted MCBs and several spatial-temporal analysis methods, this study presents a comprehensive understanding of MCB dynamics in terms of number, area, and shape, over downstream of TGD during the period 1985–2018. On average, a total of 140 MCBs were detected and grouped into four types representing small ($<2$ km$^2$), middle ($2$ km$^2 - 7$ km$^2$), large ($7$ km$^2 - 33$ km$^2$) and extra-large size ($>33$ km$^2$) MCBs, respectively. MCBs number decreased after TGD closure but most of these happened in the lower reach. The area of total MCBs experienced an increasing trend ($2.77$ km$^2$/yr, $p$-value $< 0.01$) over the last three decades. The extra-large MCBs gained the largest area increasing rate than the other sizes of MCBs. Small MCBs tended to become relatively round, whereas the others became elongate in shape after TGD operation. Impacts of TGD operation generally diminished in the longitudinal direction from TGD to Hankou and from TGD to Jiujiang for shape and area dynamics, respectively. The quantified longitudinal and temporal dynamics of MCBs across the entire Yangtze River downstream of TGD provides a crucial monitoring basis for continuous investigation of the changing mechanisms affecting the morphology of the Yangtze River system.

**Keywords:** Landsat images; remote sensing; time series analysis; Three Gorges Dam; Dam effects; Geomorphic dynamics; River bar; River island

## 1. Introduction

Most large rivers in the world are strongly regulated and engineered by artificial facilities [1]. The most common facilities are dams, as they lead to discontinuity and alteration in the hydrological and ecological processes of river systems [2,3]. Accordingly, increasing attention has been paid to the various impacts of damming on the hydrological and ecological processes of downstream systems, such as the morphodynamics of river channel forms [4,5]. One type of channel form that is potentially affected by damming is the mid-channel bar (MCB), also known as river island (i.e., an elevated region surrounded by river water) [6,7]. As an essential geomorphic unit and component in a fluvial system,

MCBs directly influence a river's morphological evolution and sediment archiving through their interaction with the flow and sediment transport [8–11]. Therefore, MCB dynamics in the context of damming intervention are especially important in maintaining the stability of river channels [12] and ecosystem functions (e.g., biodiversity and habitat providing) of the related hydro-ecosystems [13,14].

Under natural river regimes, MCBs often experience predictable geomorphic evolution in terms of area, shape, height, and location, due to longitudinal and temporal variations in flow and sediment load [15,16]. With intensive human interventions such as dam operation, natural river regimes are gradually replaced by regulated flow and sediment load, which dramatically affects the MCBs' evolution processes and thereby creates scientific and social concerns internationally [10,17,18]. For example, the dynamics of MCBs together with the changes of dam-induced channel hydrologic regime have been examined in large rivers across the globe, such as the UK [19], Australia [20], the USA [21], France [22], and China [23]. It is crucial to explore the longitudinal and temporal dynamics of MCBs to reveal the response of bar development to the associated impact factors (e.g., damming, water flow, sediment load, and boundary conditions) and, eventually, to project the MCB evolution in the future scenarios [11,24]. Therefore, understanding MCB dynamics is important not only for developing geomorphic theory but also for informing hydraulic engineering practices on rivers, particularly in large rivers that experience extensive anthropogenic influences [6,10].

As the largest river in China, the Yangtze River plays a key role in maintaining ecological security (e.g., wetland protection, fishing management, and shoreline protection) and economic development (e.g., shipping, sand supply, and land development) [25]. This is especially true for the downstream reach of the three gorges dam (TGD) on the Yangtze River as it passes through one of the country's most densely populated and most industrialized regions [26,27]. Historically, the MCBs in the mid-lower reach of the Yangtze River (i.e., downstream of TGD) are considered important land resources for channel dredging, habitat, and even agricultural development and urbanization, depending on the sizes and geographical conditions (e.g., stability, vegetation, and soil) [12]. Over the past three decades, the hydrodynamics in this reach, however, experienced extensive changes due to various anthropogenic activities, such as the construction of TGD [28]. By considering such massive environmental alterations and the ecological importance of the MCBs, two important questions have been raised: (1) How did MCB dynamics differ between the pre- and post-dam periods? (2) To what extent were they influenced by the dam?

Many previous investigations of the MCB dynamics with respect to damming or reservoir construction have been made over the last five decades. Petts [19] reviewed potential bar dynamics subsequent to dams along multiple British rivers. These earlier investigations were focused more on geomorphologic theory, such as the description of development stages (e.g., formation, migration, translation, and channel equilibrium) through experimental work or field studies, but within a limited longitudinal-temporal span [9]. Since the 1980s, research on MCB dynamics in the context of large scale and long-term temporal span, however, received increasingly wider attention [10,15], thanks to the rapid development of Remote Sensing (RS) technology and the sophisticated requirements for river regulation [29,30]. Among these studies, the common observations are that the area and number of MCBs tend to decrease due to the dam-induced sediment supply reduction [1,23,31]. In other cases, however, different patterns have been observed. In the Missouri River, Sanford [32] showed that the dam-regulated flow caused a series of consequences on bars, such as area reduction, less bar migrations, and more bar aggregation. Kiss and Balogh [33] showed that MCBs developed quickly in the upstream and laterally in the whole section of the Dráva River because of the coarse sediment supply and the decreasing stream energy due to the dam. These findings suggest that the downstream geomorphic adjustments to dam operations could result in different types of MCB dynamics (e.g., emergence, stability, or erosion), which depends upon the new hydraulic regime, sediment supply, and the type of structures employed [1,34].

Since the closure of TGD, the effects of dramatic dam operation on its downstream reaches have been discussed extensively. Most of the topics are focused on flow and sediment regime changes and

channel morphology dynamics [17,35,36]. However, only a few studies analyzed the MCB dynamics at a large spatiotemporal scale with the help of RS technology [12,23,30,37]. For example, in the reach immediately downstream of TGD, Wang et al. [23] found that the total area of MCBs extracted from Landsat images reduced dramatically by 19.23% from 2003 to 2016, accompanied with an increase of water surface width. Lou et al. [12] focused on the lower reach and found the MCBs' total area exhibited a decreasing trend in the pre-TGD period, but an increasing trend in the post-TGD period. It is noted that these results were based only on the area metric of large MCB in a limited reach. Such results are too limited to support a systematic understanding of the MCB dynamics over the entire Yangtze River downstream of TGD.

This work, therefore, made the attempted to comprehensively analyze the MCBs dynamics over the entire Yangtze River downstream of TGD throughout the last three decades (covering pre-TGD and post-TGD periods), by (1) inventorying MCBs extracted from time-series satellite imagery and (2) understanding the spatial (or longitudinal) and temporal patterns of the MCBs under the local damming and other anthropogenic contexts with three morphology-related metrics: MCB number, area, and shape.

## 2. Materials and Methods

### 2.1. Study Area

The Yangtze River originates on the Qinghai-Tibet Plateau 5100 m above sea level and flows eastward to the East China Sea. It is the largest river on the Eurasian continent and third longest river in the world, with a length of 6300 km and a drainage area of about $1.8 \times 10^6$ km$^2$. The climate is controlled by the East Asian Monsoon with the basin-wide precipitation averages around 1100 mm/yr [38]. The mainstream of the Yangtze River consists of the three reaches (Figure 1): the upper reach from the head to Yichang (YC), the middle reach from YC to Hankou (HK), and the lower reach from HK to the estuary (EST) near Shanghai City. TGD lies 43 km upstream of YC (Figure 1). It is the largest hydropower dam in the world. TGD was closed in 2003 and has been fully operational since 2010. The general principle guiding TGD operation is to keep the water level of the Three Gorges Reservoir (TGR) at the lowest (to 145 m) in flood season from May to September and at the highest (to 175 m) in dry season from October to April in the following year [39]. Accordingly, the hydrological and sediment regimes in the downstream of TGD are altered [24].

The downstream reach of TGD served as our study area (Figure 1). It is 1560 km long and consists of both middle and lower reaches of the Yangtze River. Its channel is much wider (from 900 m to 1800 m) and the river's slope is smaller (with an average slope of 0.0256‰) than these in the upstream of TGD. Surrounding the channel is one of China's most densely populated and highly industrialized regions [39]. Thus, apart from hydrology alteration by TGD operation, the downstream reach of TGD is also affected by various human activities, such as navigation, sand mining, agricultural activities, and urbanization. Moreover, the ecological values of this reach have been highly recognized, such as wetland protection, rare animal protection, and ecological sustainability maintenance [40].

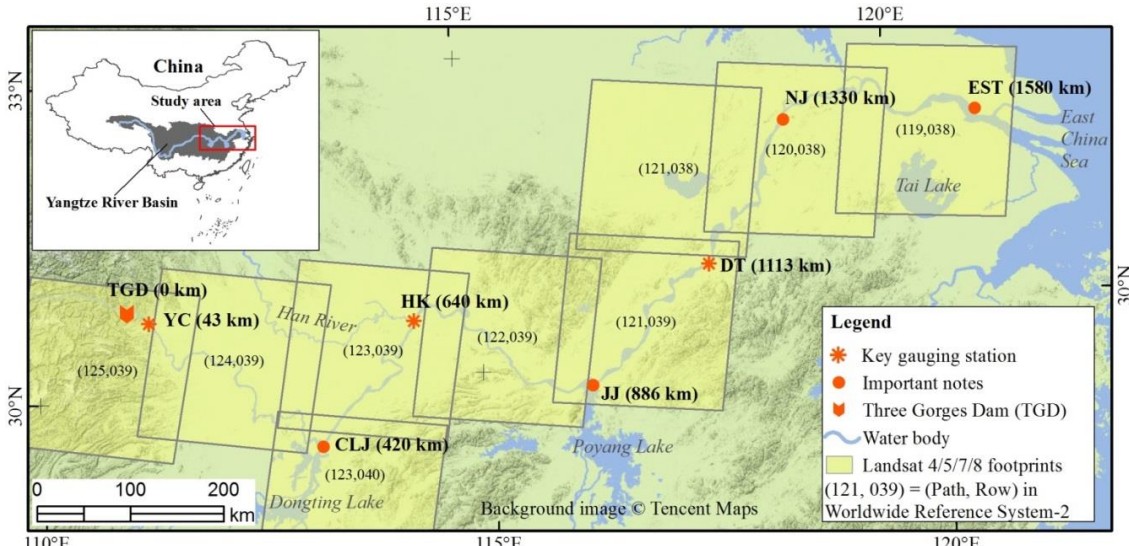

**Figure 1.** Geographic location of the study area (downstream of the Three Gorges Dam or TGD) and corresponding footprints of Landsat 4/5/7/8 images. Location names, TGD, YC, CLJ, HK, JJ, DT, NJ, and EST, are the acronym of Three Gorges Dam, Yichang, Chenglinji, Hankou, Jiujiang, Datong, Nanjing and estuary of the Yangtze River, respectively. The closed number following location name indicates the downward distance of that location from TGD. For example, "YC (43km)" means Yichang gauging station is located at 43 km downstream of TGD.

## 2.2. Data Sources

### 2.2.1. Landsat Images

The free Landsat archive images obtained from 1985 to 2018 were used as raw image sources for extracting long-term MCB datasets. They include Landsat -4/5 TM (1985–2011), -7 ETM+ SLC-ON (1999–2003), -7 ETM+ SLC-OFF (2009–2012), and -8 OLI images (2013–2018). All the images are standard Landsat surface reflectance level-2 products with 30 m spatial resolution. They were ordered from the United States Geological Survey (USGS)'s Earth Explorer (http://earthexplorer.usgs.gov/) and downloaded from the USGS's Earth Resources Observation and Science Center Science Processing Architecture on Demand Interface (https://espa.cr.usgs.gov/). Landsat-4/5 TM and Landsat-7 ETM+ surface reflectance data were produced by the software of Landsat Ecosystem Disturbance Adaptive Processing System (LEDAPS) [41]. More details about the LEDAPS algorithm and Landsat-4/5/7 Surface Reflectance data products can be found in [42]. Landsat 8 surface reflectance data were generated from the Landsat Surface Reflectance Code (LaSRC). Details of the LaSRC and Landsat 8 Surface Reflectance data products can be found in [43]. Moreover, images with the same scene (e.g., path/row = 121/039) were consistently geo-registered within prescribed image-to-image tolerances of less than 12 m, making them suitable for time-series pixel-level analysis [44].

Not all Landsat images covering the study area were suitable for the analysis. A total of 553 images were carefully selected based on two criteria. First, the cloud cover should be less than 10% of the whole scene. Second, images should be acquired during the dry season (i.e., November to March) [24] for the following reasons. The presence of MCBs is largely affected by the water level at the imaging time [45]. Therefore, the water levels around the same MCBs should be identical for different imaging times, which is important for analyzing the temporal dynamics of MCBs. However, such an ideal requirement is unrealistic for the Landsat images acquired at a frequency of 16 days, and the practical way to proceed is to keep the water levels within a relatively small range rather than identical (Figure 2a). Moreover, the water level variation in the dry season is much smaller than that in the flood season (Figure 2b), which can minimize the uncertainty of the extracted MCB caused by

water level fluctuation [46]. Generally, a 0.5 m change of water level could cause about 0.1 km$^2$ area change of MCBs (Figure 2c).

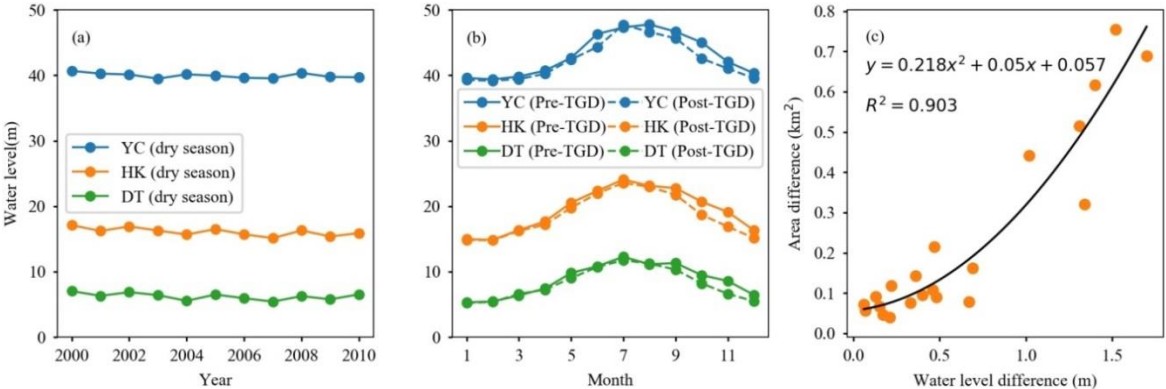

**Figure 2.** Water level variations at the three gauging stations (YC, HK, and DT as shown in Figure 1) from 2000 to 2010: (**a**) annual variations of monthly-averaged water levels in dry season (i.e., November to March) and (**b**) seasonal variations of annual-averaged water levels over pre-TGD (2000-2002) and post-TGD (2003–2010) periods. (**c**) The sensitivity of MCB area changes to water level fluctuation.

### 2.2.2. High Spatial Resolution (HSR) Images

Using Google Earth, some HSR images were carefully selected as source data to obtain validation MCBs, which were then used to assess the accuracy of MCB extracted from the Landsat images. Since the presence and extent of MCB are highly dependent on the water level that varies over time, the acquisition dates of one Landsat image and the corresponding validation HSR image should be the same. However, this ideal criterion cannot always be met in practice. Therefore, this research used a more reasonable criterion: the difference of acquisition dates between HSR images and corresponding Landsat images is limited to 7 days. Finally, 19 HRS images were selected and downloaded from Google Earth (Table 1).

**Table 1.** High Spatial Resolution (HSR) images and spatially corresponding Landsat images for accuracy assessment.

| HSR Images in Google Earth | | Landsat Images | | | |
| --- | --- | --- | --- | --- | --- |
| Source | Acquisition Date | Sensor | Path | Row | Acquisition Date |
| DigitalGlobe | 2010-12-02 | Landsat 5 | 119 | 038 | 2010-12-01 |
| CNES Airbus | 2013-12-10 | Landsat 8 | 119 | 038 | 2013-12-11 |
| DigitalGlobe | 2003-01-04 | Landsat 5 | 120 | 038 | 2003-01-07 |
| CNES Airbus | 2013-11-15 | Landsat 8 | 120 | 038 | 2013-11-20 |
| DigitalGlobe | 2014-11-18 | Landsat 8 | 120 | 038 | 2014-11-16 |
| CNES Airbus | 2017-12-12 | Landsat 8 | 120 | 038 | 2017-12-11 |
| DigitalGlobe | 2008-12-10 | Landsat 5 | 121 | 039 | 2008-12-08 |
| DigitalGlobe | 2015-02-13 | Landsat 8 | 121 | 039 | 2015-02-08 |
| DigitalGlobe | 2017-12-19 | Landsat 8 | 121 | 039 | 2017-12-21 |
| DigitalGlobe | 2017-12-26 | Landsat 8 | 122 | 039 | 2017-12-21 |
| CNES Airbus | 2018-01-11 | Landsat 8 | 122 | 039 | 2018-01-11 |
| DigitalGlobe | 2004-02-13 | Landsat 5 | 123 | 039 | 2004-02-14 |
| DigitalGlobe | 2006-12-19 | Landsat 5 | 123 | 039 | 2006-12-17 |
| DigitalGlobe | 2017-02-16 | Landsat 8 | 123 | 039 | 2017-02-13 |
| CNES Airbus | 2015-01-01 | Landsat 8 | 124 | 039 | 2014-12-31 |
| DigitalGlobe | 2016-12-05 | Landsat 8 | 124 | 039 | 2016-12-08 |
| CNES Airbus | 2017-01-22 | Landsat 8 | 124 | 039 | 2017-01-16 |
| DigitalGlobe | 2017-12-24 | Landsat 8 | 124 | 039 | 2017-12-17 |
| DigitalGlobe | 2018-01-09 | Landsat 8 | 124 | 039 | 2018-01-09 |

2.2.3. Gauged Datasets

River and suspended sediment discharges were collected at three key gauging stations (YC, HK, and DT) from the annually released Yangtze River Sediment Bulletin (http://www.cjw.gov.cn/zwzc/bmgb/). These data were yearly aggregated from daily measurements.

*2.3. Producing MCBs*

2.3.1. Extracting MCBs with Landsat Images

MCB is a land area surrounded by channel water. Therefore, the key to extracting MCB is to identify "holes" in channel water maps that can be produced from Landsat images. The procedure consists of five main steps as illustrated in Figure 3 and detailed below.

- Step 1: Clipping image

For each Landsat scene, the Yangtze River channel only occupies a small portion of the image, thus, it is necessary to clip out the small portion containing the Yangtze River channel from the entire Landsat scene to improve computing efficiency. The clipper is a 1-km buffered Yangtze River channel polygon which was manually delineated from Landsat 8 images (Figure 1).

- Step 2: Calculating water index

Landsat water indices have widely been used for mapping surface water bodies [47], for example, the normalized difference water index [48], the modified normalized difference water index (MNDWI) [49], and the automated water extraction index for images with and without considering shadows [50]. Previous studies have suggested that the MNDWI performs well in various conditions when compared to other water indices [50–53]. This study, therefore, adopted MNDWI. Equation (1) was applied to all the clipped satellite images to generate MNDWI images using a Python script with the ArcGIS software environment (version 10.3.0.4322, ESRI, Redlands, CA, USA) [54].

$$\text{MNDWI} = \frac{\rho_{\text{green}} - \rho_{\text{swir1}}}{\rho_{\text{green}} + \rho_{\text{swir1}}} \tag{1}$$

where $\rho_{\text{green}}$ and $\rho_{\text{swir1}}$ are surface reflectance in the green band and the short-wave infrared band (i.e., band 5 for Landsat-4/5/7 and band 6 for Landsat 8), respectively.

- Step 3: Determining the optimal threshold

To classify channel water from the land area, an optimal threshold is needed to segment the MNDWI images. However, the optimal threshold often varies with space and time [52]. Since there are hundreds of MNDWI images, it is crucial to automatically determine the optimal threshold of each MNDWI image independently. The modified histogram bimodal method (MHBM) developed by Zhang et al. [53] was applied to determine the optimal threshold of each image. 548 out of 553 MNDWI images featured distinct bimodal histogram patterns which were very suitable for the application of MHBM. For the other five MNDWI images with non-bimodal histograms, the MCBs were extracted manually.

- Step 4: Classifying water and non-water areas

For an MNDWI image, pixels with values larger than the determined threshold were classified as water areas the others were classified as non-water areas.

- Step 5: Extracting MCBs

First, the raster format water body area were converted into vector shapefile polygons with smoothed edges. Next, the holes in the vector water map were converted to polygons that were considered as candidate MCBs. Geographically, all the holes contained in the channel water are MCBs. For the Yangtze River, however, some of the holes can also be large ships because this river is one of the busiest inland waterways in the world [55]. Such detected ships are often less than 0.02 km², as confirmed by carefully inspecting the high spatial resolution images provided by Google Earth. Finally, the "true" MCBs were selected from the candidate MCBs with an area larger than 0.02 km².

All these processes were done by specific Python scripts in the ArcGIS environment (version 10.3.0.4322, ESRI, Redlands, CA, USA) [54].

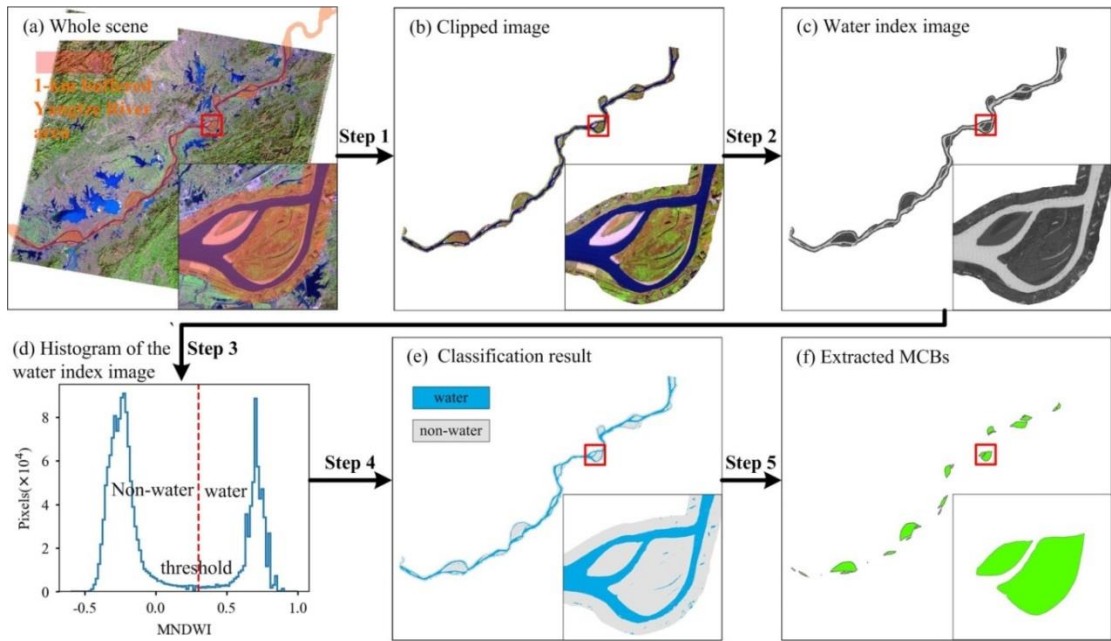

**Figure 3.** Main steps for automatic extraction of MCBs. (**a**) The whole scene of Landsat-5 image in false color (R: Band 5, G: Band 4, and B: Band 3) acquired, for example on 14 January 2010 (path=121, row=039). (**b**) Clipped image by the 1-km buffered Yangtze River polygon GIS data. (**c**) Water index image generated by the algorithm of the modified normalized difference water index [49]. (**d**) Histogram of the water index image for determining the optimal threshold. (**e**) Raw classification result by applying the threshold method. (**f**) Extracted MCBs from the raw classification result. Steps 1–5 are detailed above.

### 2.3.2. Calculating MCBs' Attributions

(1)    Unique Coding

Each MCB generated from a Landsat scene was initially assigned a unique identification code. The code consists of three parts: the scene's path and row numbers, the scene's acquisition date, and the inherent geometry identification number of the MCB. For example, code "12103920100114005" means the corresponding MCB is identified as "005" in the scene which was obtained at path "121" and row "039" on date 14 January 2010. Therefore, each MCB had its own unique code, which could complicate the dynamic analysis of the MCB in the programming process. To avoid this inconvenience, the MCBs located at the same place but generated from different Landsat scenes with different acquisition dates should share the same code. In this study, the code that contains the earliest acquisition date was assigned to all the MCBs at the same location. The unique codes of all MCBs were listed in the supplementary Table S1.

(2)    Area and shape index

Our analysis focused on area and shape attributes. Since MCBs often in transverse or lobate shapes [9], the length-width ratio (LWR, Equation (2)) may serve as an appropriate metric to indicate an MCB's shape and its dynamics caused by deposition or erosion:

$$LWR = \frac{L}{W} \tag{2}$$

where $L$ and $W$ represent the length and width of the convex rectangle of an MCB, respectively. Both MCB's area and convex rectangle were generated with the ArcGIS environment (version 10.3.0.4322,

ESRI, Redlands, CA, USA) [54]. The area and LWR values of all MCBs were listed in the supplementary Table S1.

### 2.4. Processing Validation Data

The validation MCBs that occurred in the HSR images (Table 1) were manually delineated using ArcMap. The same MCBs but derived from Landsat images (Table 1) for accuracy assessment were also selected. In total, 49 pairs of MCBs (validation MCBs vs. assessment MCBs) were matched (see supplementary .kml files). Both their area and LWR values were calculated and were used for further accuracy assessment analysis.

### 2.5. Analysis Methods

Many geographical and environmental temporally dynamic phenomena experience structural breaks which could result from some sudden events, such as disasters or human intervention. Previous case studies have reported that the temporal dynamics of some MCBs might have been affected by the closure of TGD [12,28]. In our study, these temporal structural breaks were tested systematically. The Chow test [56] used by many other researchers [57,58] was applied to test TGD induced structural break. For a time series of MCBs data, the Chow test examined the null hypothesis: H0 (there is no structural change) vs. the alternative hypothesis H1 (there is a structural change). The test was conducted by running three separate regressions and an *F*-test: (1) a linear regression with the entire time series data; (2) two separate linear regressions with data before (Pre-TGD) and after TGD closure in 2003 (Post-TGD); and (3) an *F*-test to determine whether a single linear regression is more efficient than two separate regressions. If this *F*-test was significant ($p$-value $< 0.05$), the null hypothesis (H0) was rejected in favor of the alternative hypothesis (H1) was accepted. This process was applied to each individual MCB and grouped MCBs (as detailed in Section 3.2.2) in the R software environment [59].

## 3. Results

### 3.1. Understanding of the Extracted MCBs

#### 3.1.1. Accuracy of Extracted MCBs Data

Accuracy of the MCBs extracted from Landsat images was assessed using the validation MCBs generated from HSR images. These validation MCBs were distributed along the entire study area (Table 1) and represent a wide range of MCB characteristics both in area (size) and shape (LWR) (Figure 4). For both area and shape, the Landsat derived MCBs are consistent with the validation MCBs. The Root Mean Square Error (RMSE) values for area and LRW were 0.25 km$^2$ and 0.22, respectively. These results show that the accuracy of the procedures for extracting MCBs with Landsat images is acceptable and that the extracted MCBs are reliable for performing area and LWR dynamic analysis.

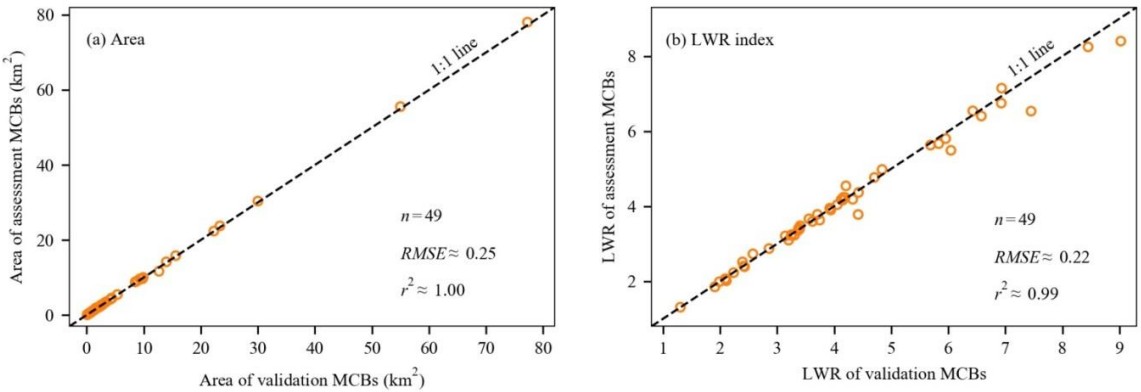

**Figure 4.** Accuracy assessment of MCBs for area (**a**) and LWR index (**b**).

### 3.1.2. General Overview of MCBs

This study provides the first opportunity for examining the general pattern of MCBs downstream of TGD. The individual MCB area varies from 0.08 km$^2$ to 223 km$^2$ (Figure 5). Moreover, the area histogram illustrates a strong right-skewed distribution as shown in Figure 5a-1,a-2. Specifically, half of the MCBs are less than 2 km$^2$, 25% of MCBs are within an area range from 2 km$^2$ to 7 km$^2$; 20% of MCBs within an area range from 7 km$^2$ to 33 km$^2$; and only 5% of MCBs are larger than 33 km$^2$.

**Table 2.** Four different types of MCBs.

| Type | Explanation | Area range (km$^2$) Minimum | Area range (km$^2$) Maximum | Proportion |
|------|-------------|---------|---------|------------|
| T1 | Small size | 0.02 [a] | 2 | 50% |
| T2 | Middle size | 2 | 7 | 25% |
| T3 | Large size | 7 | 33 | 20% |
| T4 | Extra-large size | 33 | 223 [b] | 5% |

Note: [a] Only MCBs with their area larger than 0.02 km$^2$ were considered as "true" MCBs as described in Step 5 in extracting MCBs. [b] 223 km$^2$ was the largest area among the MCBs' area data.

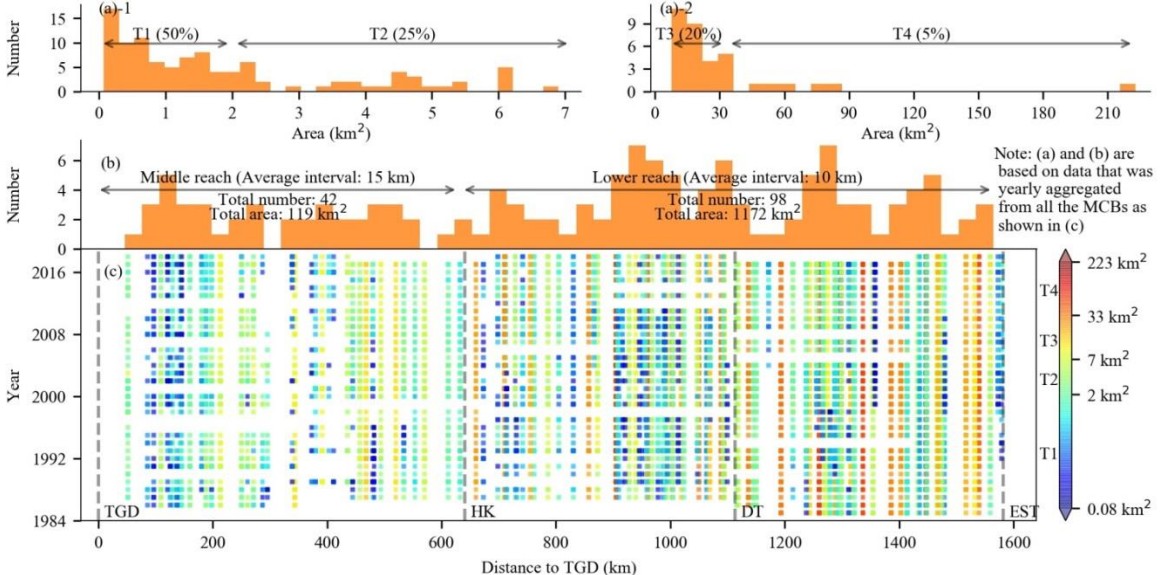

**Figure 5.** Overview patterns of MCBs extracted from Landsat images. Figures (a)-1 and (a)-2 are area histograms of MCBs with consecutive area ranges. (b) Distance-to-TGD histogram of MCBs, and (c) Longitudinal and temporal distribution of all MCBs (each dot represents a single MCB) with different sizes as indicated by their colors. T1, T2, T3, and T4 stand for four different MCB types as explained in Table 2.

MCBs area is a comprehensive indicator of both geographical processes (e.g., geomorphology evolution stage, vegetation status, and stability) and anthropological processes (e.g., land use type, development history, and sand mining) [8,15]. Thus, the observed large variations in area between MCBs indicate that the dynamic analysis of MCBs should fully consider the potential influences caused by the effects of size or scale. It is necessary to group MCBs into different size types based on specific area criteria. To the best of our knowledge, however, there are no such criteria that can be referenced. Most of the previous studies focused on large MCBs which have relatively little area variation [12,30]. In this study, considering both the area histogram distribution (Figure 5a-1,a-2) and sample size requirements for further statistical analysis, MCBs were grouped into four size types, namely T1, T2, T3, and T4 (Table 2).

## 3.2. Spatiotemporal Dynamics of MCBs

### 3.2.1. Longitudinal Distribution

The longitudinal distribution of MCBs is illustrated in Figure 5. The distribution pattern is uneven (Figure 5c). In detail, distinct distribution patterns of MCBs were observed in middle reach (TGD-HK) and lower reach (HK-EST). In the middle reach, there are 42 MCBs (30% of the total number) scattered along the channel with an average interval of 15 km from 1985 to 2018. The sum of annual averaged MCBs area in the middle reach was 119 km$^2$, accounting for 9% of the total MCBs area in the entire Yangtze River downstream of TGD (Figure 5b). By contrast, there were 98 MCBs (70% of total MCBs number) along the lower reach distributed with an average interval of 10 km, relatively denser than in the middle reach. The sum of annual averaged MCB area in lower reach was 1172 km$^2$, accounting for 91% of the total MCBs area. These distinct longitudinal distribution patterns highlight the importance of the lower reach in the analysis of MCBs from both quality and quantity perspectives.

To be more specific, the MCBs extracted in 2016 were taken as an example to illustrate the longitudinal distribution patterns of MCBs (Figure 6). It is noteworthy that the T1 MCBs (i.e., small-size) scattered along the entire reach without obvious concentration. Most of the T2 and T3 MCBs (i.e., middle and large sizes) occurred in the lower reach. All of the T4 MCBs (i.e., extra-large size) were distributed in the lowest reach, i.e., JJ-EST (Figure 6a). In other words, MCBs with large area tended to be located farther downstream from TGD, especially in the reach JJ-EST.

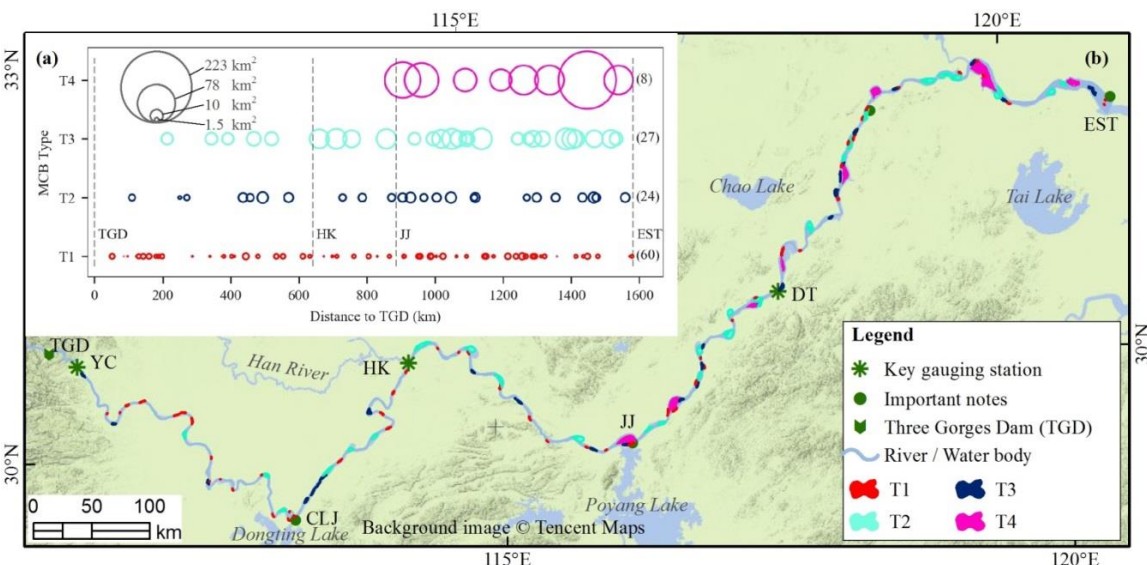

**Figure 6.** Longitudinal distribution of MCBs along downstream of TGD in 2016.

### 3.2.2. Temporal Dynamics of MCBs

The temporal dynamics of MCBs in terms of number, area, and LWR indices are described below. For each index, this study performed analyses with and without consideration of MCB type. For each temporal trend analysis, whether the closure of TGD can be considered as a structural change during the whole period was also tested. If it did, the pre-TGD, post-TGD, and the overall regressions were applied; otherwise, only the overall regression was applied.

(1)　Number

There was no statistically significant change in MCB number during the whole study period as shown in Figure 7a ($a = -0.08$, $p$-value $> 0.05$). However, TGD closure year (in 2003) was detected as a structural change and different trends were observed between the pre-TGD and post-TGD periods. In the pre-TGD period, MCB number showed an increasing trend ($a = 0.4$ and $p$-value $< 0.01$). Such a trend was reversed in the post-TGD period in which a significant decreasing trend was detected

($a = -0.5$, *p*-value < 0.01). To further understand these changes, trends of the four MCB types were checked as illustrated in Figure 7b–e. The number of T1 MCBs showed very similar trends as the total MCBs and the number in other three MCB types remained stable during the study period, suggesting that the phenomenon of MCB's "disappearance" or "appearance" existed in the study periods and only happened for the small size MCBs (T1). It should be noted that the so-called "disappearance" or "appearance" does not mean an MCB completely disappeared or appeared. This is because only the candidate MCBs with an area larger than 0.02 km² were detected (as explained in Section 2.3.1) and applied in our analysis. Therefore, in some cases, the "disappearance" of one MCB may be caused by flow erosion which reduced its area less than our detection criterion (0.02 km²).

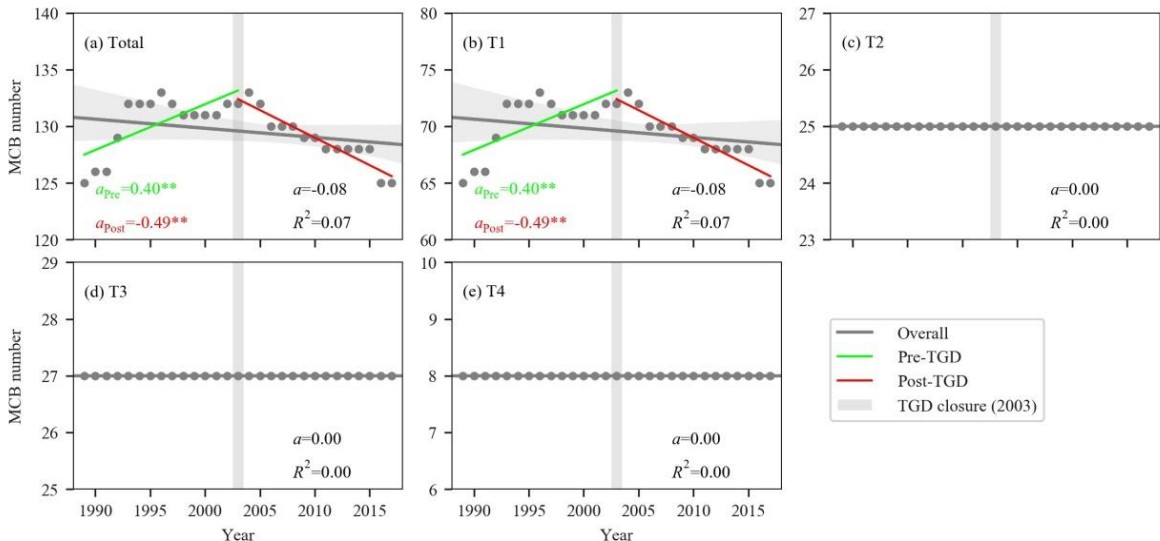

**Figure 7.** Temporal dynamics of MCB number for different groups of MCBs: (**a**) Total MCBs, (**b**) T1 MCBs, (**c**) T2 MCBs, (**d**) T3 MCBs, and (**e**) T4 MCBs. The regression parameters a and R2 stand for regression slope and coefficient of determination, respectively. Slope value with "*" and "**" indicate the corresponding *p*-value < 0.05 and < 0.01, respectively.

Since the number change only occurred on T1 MCBs, where and when did these events happen is another question. As shown in Figure 8, the appearance of new MCBs mainly happened in the pre-TGD period. The locations of those events were concentrated in the JJ-EST reach. The MCB disappearance started in 1996 and mainly occurred in the JJ-EST reach as well. It is unexpected that the immediate downstream reach of TGD (i.e., TGD-HK) only experienced two events of MCB disappearance after the closure of TGD in 2003.

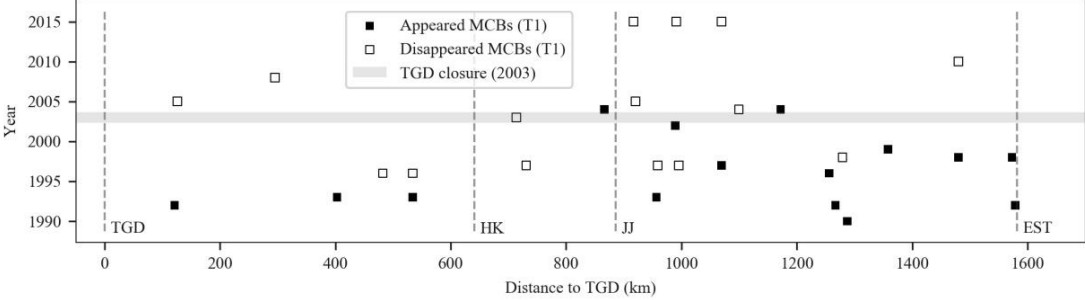

**Figure 8.** Longitudinal-temporal distribution of number change events.

(2)　Area

Overall, the area of all MCBs experienced an increasing trend with a slope of 2.77 km²/yr (*p*-value < 0.01, Figure 9a). The area of the four MCB types exhibited increasing trends (the trend for T3 is not statistically significant). The slopes of T1, T2, and T4 increased accordingly (Figure 9), from 0.53 km²/yr (*p*-value < 0.01) to 0.61 km²/yr (*p*-value < 0.01) to 1.29 km²/yr (*p*-value < 0.01). It is also noteworthy that the *Coefficient of Variation (CV)* values show a decreasing pattern from T1 to T4. The combination of these observations shows that larger MCBs tend to have higher rates of area increase (i.e., more sediment deposition) but with less area variation (i.e., more stability).

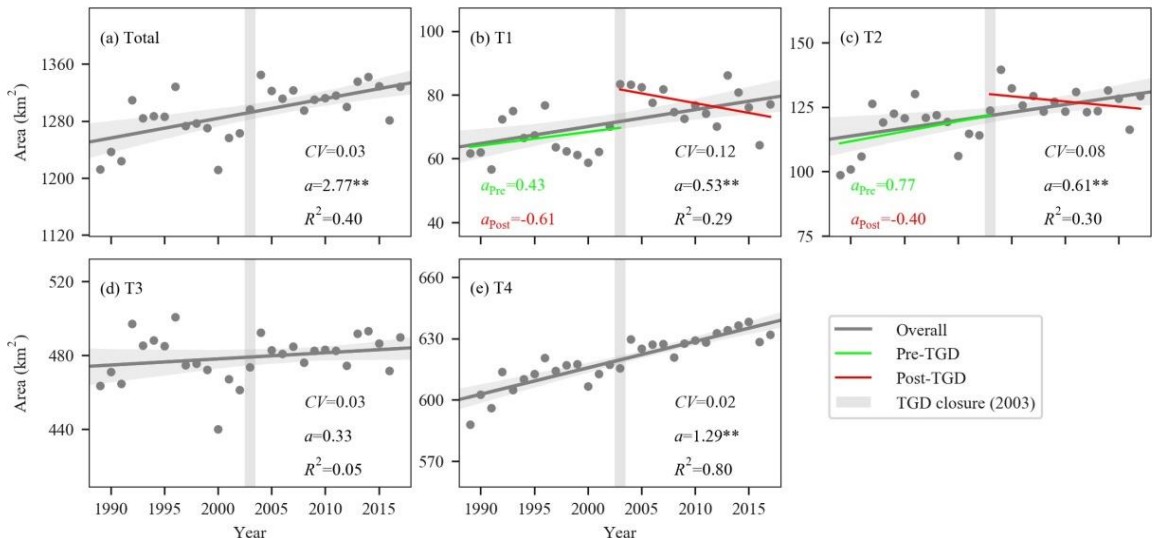

**Figure 9.** Area temporal dynamics of different groups of MCBs: (**a**) Total MCBs, (**b**) T1 MCBs, (**c**) T2 MCBs, (**d**) T3 MCBs, and (**e**) T4 MCBs. The regression parameters a and $R^2$ stand for regression slope and coefficient of determination. Slope values with "*" and "**" indicate the corresponding *p*-value < 0.05 and < 0.01, respectively. The *CV* stands for coefficient of variation which is defined as the ratio of the standard deviation to the mean and here it indicates the extent of variability in relation to the mean area of the MCBs.

As for TGD effects on MCB area dynamics, structural changes were detected for T1 and T2 MCBs, but not for the T3 and T4 MCBs (Figure 9). For T1 and T2 MCBs, their area showed increasing trends in the pre-TGD period but showed slightly decreasing trends post-TGD. This result suggests that the closure of TGD might have a significant impact on area dynamics of small (T1) and middle (T2) size MCBs. In other words, smaller size MCBs are more likely to be influenced by the dam operation and experience more sediment erosion, whereas larger size MCBs probably have more resistance to the impact of dam operation.

(3)　LWR shape index

Different types of MCBs had distinct basic shape characteristics as shown in Figure 10. Generally, the larger size MCBs have smaller LWR values (i.e., *μ* values in Figure 10), namely, 4.97 (T1) > 4.12 (T2) > 3.16 (T3) > 2.24(T4). According to the definition of LWR, this observation illustrates that the relatively small size MCBs (e.g., T1 and T2) tended to elongate, whereas relatively large size MCBs (e.g., T3 and T4) tended to appear more rounded in shape.

As for the overall temporal dynamics, it is noted that the LWR of the total MCBs showed a significant decreasing trend in the whole study period (*a* = −0.02, *p*-value < 0.01, Figure 10a), indicating that the shape of MCBs tended to change from elongate to relatively round. As for the different types of MCBs, their shape temporal dynamics illustrated distinct trends. T1 and T2 MCBs showed significant decreasing trends, while T3 and T4 MCBs showed a significant increasing trend in the whole period (Figure 10b–e). These results show that the T1 and T2 MCBs were gradually

becoming relatively round whereas the T3 and T4 MCBs tended to become more elongate from 1989 to 2017. Apart from their change direction, their change rates also showed an interesting pattern: the larger size MCBs, the slower change rate of LWR dynamics, i.e., 0.03 /yr (T1) > 0.02 /yr (T2) > 0.01 /yr (T3) > 0.001 (T4). This pattern indicates that the shapes of T3 and T4 MCBs were more stable than those of T1 and T2 MCBs.

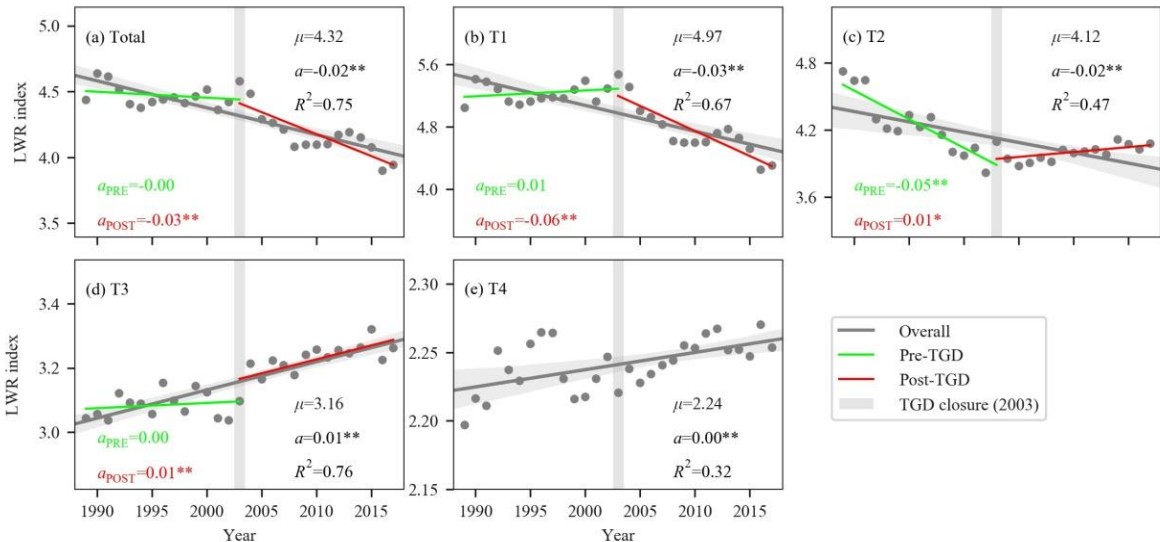

**Figure 10.** LWR temporal dynamics of different groups of MCBs: (**a**) Total MCBs, (**b**) T1 MCBs, (**c**) T2 MCBs, (**d**) T3 MCBs, and (**e**) T4 MCBs. The regression parameters a and $R^2$ stand for regression slope and coefficient of determination. Slope values with "*" and "**" indicate the corresponding *p*-value < 0.05 and < 0.01, respectively. Parameter μ stands for the mean value of LWR.

Structural changes were also detected in the LWR dynamics of T1, T2, and T3 MCBs at TGD closure year. For T1 MCBs (Figure 10b), they experienced an insignificant trend change in the pre-TGD period and a significantly decreasing trend in the post-TGD period. Both T2 and T3 MCBs showed a significant increasing trend in the post-TGD period (Figure 10c,d).

## 4. Discussion

### 4.1. Scale Effects on the Temporal Dynamics

The temporal dynamics of MCBs indicate that different MCB types had distinct trend characteristics. More specifically, the large MCBs seem to have more stability than the relatively small ones in terms of variations in both area and LWR. In other words, the size of MCBs may influence their temporal dynamics, which suggests the presence of scale effects. Figure 11 illustrates the scale effects of MCBs on the temporal stabilities of area and LWR in the overall, pre-TGD, and post-TGD periods, respectively. As for the overall period, the *CV* of area and LWR were both decreasing (becoming less variable and more stable) as the size or scale of the MCBs increased. Such a pattern also applies to the pre-TGD, and post-TGD periods. These results imply that the larger the size of MCB, the greater relative stability in area and shape variations, whereas the smaller size of the MCBs, the less stable in terms of area and shape variations. In previous studies, only one specific scale MCBs were focused (such as large-size MCBs), and the temporal dynamic patterns of other sizes of MCBs were not addressed [23,30].

Three reasons could explain the observed scale effect. The first is the growth of vegetation. Compared to small MCBs, large MCBs are more likely to be covered by vegetation [60,61]. In turn, the vegetation would provide more resistance to external forces such as sediment erosion [15,62]. The second is the strength of human interventions. Large size MCBs are more likely to have been occupied and modified by human beings and thus received more protection that can, in turn, enhance their

stability. For example, people would harden MCB banks to prevent them from being eroded, especially on agriculturalized or industrialized MCBs in the lower reach [63]. The third is the effects of flooding. Generally, the small size MCBs have short evolution histories with lower height compared to the large size MCBs. Therefore, they are more likely to be inundated in a flooding event and to experience large variations in sediment erosion or deposition [64]. In contrast, large size MCBs would experience relatively fewer impacts by such flooding events.

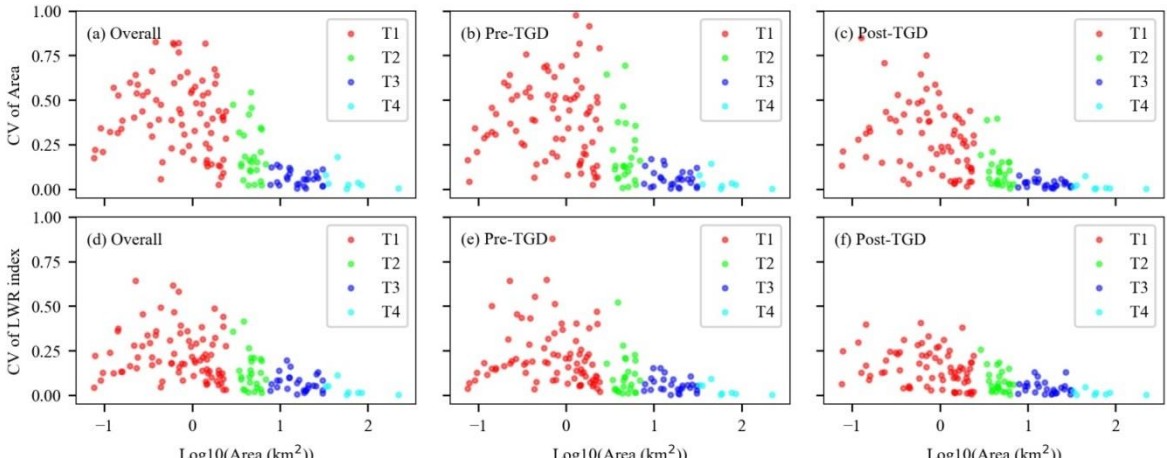

**Figure 11.** *CV*-indicated temporal stability of all individual MCBs changed with their sizes (indicated by log-transformed area) in different periods (overall, pre-TGD, and post-TGD).

### 4.2. Potential Effects of TGD Operation

It has been widely acknowledged that MCBs can be heavily influenced by flow and sediment regimes [31,62]. Downstream of TGD, marked changes of flow and sediment regimes between the pre- and post-TGD periods have been observed [39,65]. Although many previous studies focused on different periods, some common themes can be identified. First, there is no significant change in the annual water discharge between pre-TGD and post-TGD periods in the last three decades [24] (Figure 12a). However, the peak river discharge during the flood season was reduced due to the operation of TGD (Figure 12c). Second, the amount of sediment carried by the river started to decrease in the 1990s, but it dropped dramatically after TGD closure in 2003 (Figure 12b) [66]. Third, the sediment carrying capacity decreased in the lower reach [12,67]. The direct result of these changes is that the water immediately downstream of TGD is in a sediment-hungry condition, which would carry away some sediment in the riverbed or in the submerged parts of MCBs [68].

The operation of TGD could have impacted the temporal dynamics of MCBs in the downstream reach. A further question that needs to be discussed is whether these influences vary spatially as a function of the distance of MCBs to TGD. The frequency of structural changes (FSC) is proposed here as an indicator for analyzing this question. The FSC is identified as the frequency of temporal structural changed MCBs among every 10-MCBs bin. The temporal structural changes were tested for both area and shape metrics between pre-TGD and post-TGD periods (see Section 2.5). For a bin, the 10 MCBs are counted in ascending order by their distances to TGD. Therefore, the higher the FSC, the more likely the structural changes were caused by TGD operation (Figure 13a).

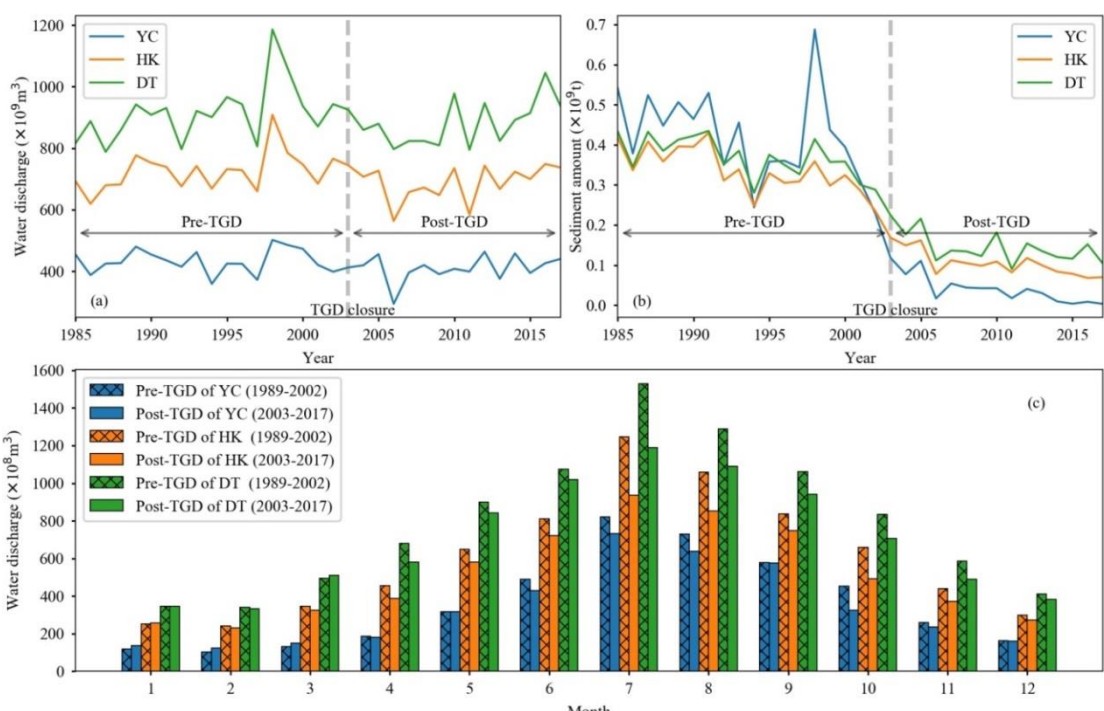

**Figure 12.** Flow and sediment regimes in the pre-TGD and post-TGD periods in the three key gauging stations. (**a**) Annual water discharge from 1985 to 2017, (**b**) Annual sediment amount from 1985 to 2017, and (**c**) Monthly averaged water discharge from 1989 to 2002 (Pre-TGD) and from 2003 to 2017 (Post-TGD).

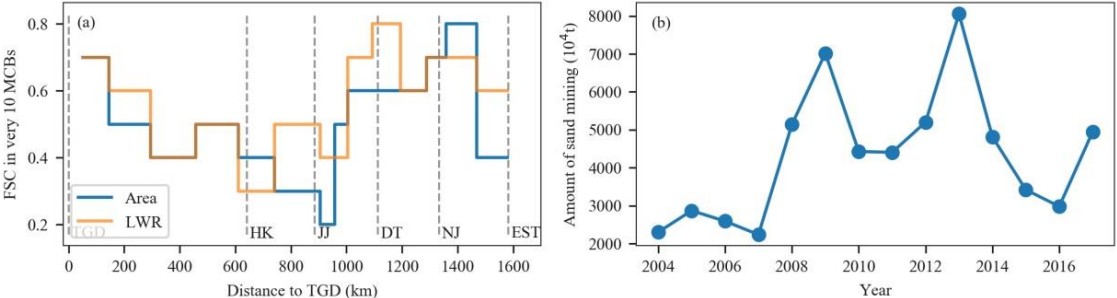

**Figure 13.** (**a**) Frequency of structural changes (FSC) with distance to TGD. (**b**) The total amount of sand mining in the downstream of TGD.

For area temporal dynamics, the FSC gradually decreased from 0.7 to 0.2 in TGD-JJ reach, and then slowly increased from 0.2 to 0.8 in the JJ-NJ reach (Figure 13a). Similarly, the FSC of LWR gradually decreased from 0.7 to 0.3 in TGD-HK reach, and then slowly increased from 0.3 to 0.8 in the HK-DT reach. The decreasing patterns of FSC indicate that, to some extent, the effects of TGD operation decreased as the distance to TGD increased. The minimum effects of TGD occurred at HK and JJ for the LWR dynamics and area dynamics, respectively. This result is similar to observations made in previous studies that focused on the effects of TGD on the water regime and channel dynamics [46,69]. For example, [46] found that the operation of TGD reduced the water level in the river downstream, but the effects were most pronounced immediately downstream of the dam and generally diminished in a longitudinal direction from TGD to the estuary. As for the channel dynamics, Yuan et al., [69] reported that channel scour was found in the post-TGD period and was strongest immediately downstream of the dam and then decreased longitudinally. In contrast, this study noted an unexpected increasing

trend of FSC in the HK-DT and JJ-NJ reaches for the LWR and area dynamics, respectively. Do these results imply the TGD effects increased in these lower reaches after the effects had been considered minor as discussed just above? It may be unreasonable to make such a judgment and thus more evidence is needed to explain the increasing trend we observed in this study. In fact, apart from the aforementioned hydrological factors, some human activities, such as sand mining, may also play an important role in changing the MCB area and LWR dynamics [16], especially for the middle- and small-size MCBs in the lower reach like JJ-DT. Before 2003, sand mining activity was banned. However, after carrying out the Regulation of Sand Mining Management in the Yangtze River in June 2003, the amount of annual sand mining increased dramatically (Figure 13b). Those sand mining activities that were reported mainly in the lower reach, particularly in the JJ-EST reach [70]. Since the large-scale sand mining activities often happened near MCBs [71], it could cause a structural change of MCBs dynamics after 2003. Therefore, the causes for MCBs' dynamics in the lower reach (JJ-EST) is more complicated than that in the reach immediately downstream of TGD (TGD-HK) and more supporting data and quantitative analysis are needed for the further evaluation of TGD effects on the lower reach.

In addition to the longitudinal extent of TGD effects, this study was also interested in how the TGD effects were expressed. Figure 14 indicates that the majority of MCBs experiencing a structural change habited an opposite trend in the post-TGD period compared to the pre-TGD period. Specifically, 72 out of 140 MCBs showed PR structural change (positive in pre-TGD and negative in post-TGD) in area temporal dynamics and 79 out of 140 MCBs experienced NR structural change (negative in pre-TGD and positive in post-TGD) in LWR temporal dynamics. This observation suggests that the operation of TGD could be the driving force that caused over half of MCBs to experience erosion (decreasing area) and became elongate (increasing LWR) in the post-TGD period. Thus, the operation of TGD could have significant impacts on the dynamics of MCBs in the Yangtze River downstream.

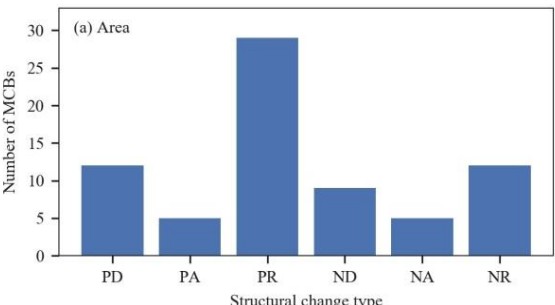 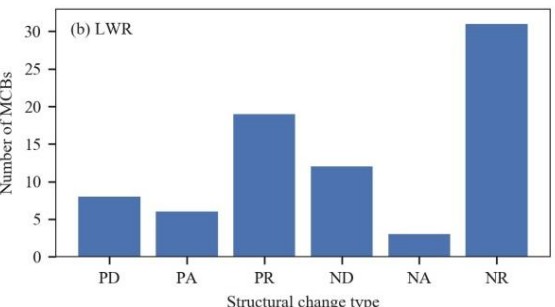

**Figure 14.** Different structural change types (details can be found in supplementary file Table S2) and their corresponding MCBs numbers for temporal dynamics of (**a**) area and (**b**) LWR. For the terms PD, PA, PV, ND, NA, and NV, the first letter "P" or "N" stands for positive trend or negative trend in the pre-TGD period; the second letter "D", "A" or "R" in each term indicates in a decelerating trend, accelerate, or a reverse trend in the post-TGD period, respectively. For example, "PR" in Figure (**a**) grouped the MCBs that have a structural change in the area temporal dynamics with positive trend pre-TGD and negative trend post-TGD.

*4.3. Limitation and Future Research*

Similar to many studies, there are several limitations and open issues associated with our current research. The area and shape of image-extracted MCBs are influenced by the water level variation due to the different image acquisition dates as demonstrated in Figure 2c. Although this unwelcome factor has been carefully considered with our best effort during the data preparation, some uncertainties could still be introduced to the temporal dynamic analyses of area and LWR. In the future studies, two further improvements can be made to minimize the impacts of such uncertainties. First, the temporal dynamics analysis should focus on a group of MCBs rather than individual MCBs and the conclusions should be made from a statistical point of view [72]. Second, future research can

use new high-spatiotemporal-resolution RS images, such as PlanetScope [73], which can enable greater consistency of water levels on different acquisition dates. The second improvement is highly recommended in the future MCBs monitoring works as these data can be obtained more economically in the near future.

Due to the relatively coarse spatial resolution of Landsat images (30 m) and the complex conditions of the Yangtze River, this study used 0.02 km$^2$ as an area threshold to determine the MCBs, i.e., only the image-extracted patches larger than 0.02 km$^2$ were considered as MCBs. As a result, some MCBs less than 0.02 km$^2$ were excluded in the analysis. In the future study, however, these tiny MCBs should also be manually checked by experts or by using high spatial resolution images to pick them out of the real non-MCBs objects, such as large ships and some man-made features.

In a real riverine system, MCBs not only experience area and shape dynamics in two-dimensional (2D) space but also experience volume and location dynamics in three-dimensional (3D) space [6,16]. To fully understand MCB dynamics and the effects of TGD, more work needs to be conducted in monitoring MCB location migrating and sand volume change. Such work will increase our understanding of sediment transportation and water management in the Yangtze River [39].

## 5. Conclusions

This study presents a systematic analysis of MCBs in the entire Yangtze River downstream of TGD in the last three decades. It gives us the first opportunity to comprehensively understand the longitudinal and temporal dynamics of MCBs during pre- and post-TGD periods. Most of the MCBs in terms of number (98 out of 140) and total area (1172 km$^2$ out of 1291 km$^2$) were scattered in the lower reach (HK-EST) with an average interval of 10 km along the channel. The temporal dynamics patterns were examined using annual MCB data using a statistical classification system. This classification system grouped the extracted 140 MCBs into four general size-types based on their area distribution. The four size-types are T1 small (area < 2km$^2$), T2 middle (area between 2 km$^2$ and 7 km$^2$), T3 large (area between 7 km$^2$ and 33 km$^2$), and T4 extra-large (area > 33 km$^2$) MCBs. For each type, the MCB temporal dynamics in total number, area, and shape index (i.e., LWR) were comparatively analyzed during the before and after TGD operation periods.

Overall, the number of MCBs increased before TGD operation and then declined significantly after the TGD operation. As for the different MCB types, only the T1 MCBs experienced such a change in number and most of this change happened in the lower reach. Although the area of all types of MCB showed an overall increasing trend, large size MCBs tended to experience larger change rate and less variation than that of the small size MCBs. Large size MCBs seemed to experience fewer impacts of TGD on their area dynamics whereas the small size MCBs likely were more influenced by the TGD operation. As for the shape dynamics, small size and middle size MCBs tended to become relatively more rounded in shape whereas the large and extra-large size MCB tended to become more elongate. Similarly, the shape dynamics of the large MCBs were more stable than those of small MCBs. This study implies that the scale (size) effects of MCBs on their temporal dynamics need to be paid more attention in future MCB analysis or in the practical management of MCBs such as channel dredging.

The operation of TGD could have significant effects on MCB dynamics. This study showed that the strength of such effects decreased as the distance to TGD increased, and minimized at HK (for LWR dynamics) or JJ (area dynamics). In contrast, the driving forces of the MCB dynamics in the lowest JJ-EST reach are more complex as more external influences such as sand mining activities are observed in the area and additional analyses are needed in the future.

**Supplementary Materials:** The following are available online at http://www.mdpi.com/2072-4292/12/3/409/s1, Table S1, Table S2, kml files.

**Author Contributions:** Conceptualization, Z.W. and S.W.; methodology, Z.W. and C.Z.; formal analysis, Z.W.; investigation, Z.W.; resources, Z.W. and G.S.; data curation, Z.W.; writing—original draft preparation, Z.W.; writing—review and editing, H.Y., C.Z., and G.S.; visualization, Z.W.; supervision, H.Y. and S.W.; funding acquisition, Z.W. and H.Y. All authors have read and agree to the published version of the manuscript.

**Funding:** This research was jointly funded by the National Natural Science Foundation of China (No: 41501096 and 51779241), Open Fund of State Laboratory of Information Engineering in Surveying, Mapping and Remote Sensing, Wuhan University (No: 18R07), and the Open Fund of State Key Laboratory of Lake Science and Environment, Chinese Academy of Sciences (No: 2018SKL006).

**Acknowledgments:** The authors would like to thank the reviewers for valuable input that increased the quality of this paper.

**Conflicts of Interest:** The authors declare no conflict of interest.

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
