# Peer review of "Remotely Sensed Mid-Channel Bar Dynamics in Downstream of the Three Gorges Dam, China"

_remotesensing, doi:10.3390/rs12030409_

Round 1
Reviewer 1 Report
This manuscript describes the spatial and temporal dynamics of mid-channel bars along the Yangtze River downstream of Three Gorges Dam, a topic of great interest to geomorphologists and resource managers since this is the world's largest dam. The study is very extensive and, importantly, based on data over a 32-year time period and the entire 1600+ km of river below the dam to the estuary. The paper thus provides some important insight beyond previous studies that have not considered such a large area and long time period. The analysis is fairly straightforward and reasonably well described, but extensive English editing will be required before the paper can be published. I have made a large number of such edits that the authors will need to take care to incorporate in a revised manuscript, but these are mostly minor wording changes. I also added a few more substantive comments in the attached PDF, to which you can refer for the details, but here are the key points:
Cite Ham and Church (2000) in the context of the effects of water level on bar area.
In Figure 2, are these points the average of the dry season months for each year? Please clarify in the text and the caption.
Use numbered citation format throughout
Line 267: Don't you want to preserve the date information? The way you've described this process, it sounds like you're throwing out the date, which would prevent any kind of change analysis, so I'm confused. Please clarify how these codes can support your time series analysis.
Figure 12 doesn't add much, or at least not as much as it could, so please elaborate on what you're trying to show.
Make sections 3.4 and 3.5 into a separate discussion section
Line 494: What do you mean by a structural change in this context? This is an important point that needs to be clarified because, at present, this analysis is not very insightful without a clear description of what constitutes a structural change.
With attention to these comments and correction of all the minor edits, this paper could become a solid contribution to Remote Sensing.

Reviewer 2 Report
Dear authors,
My first impulse was to reject from the start to review this manuscript because I discovered 75% plagiarism. Then I saw ”the history” of this manuscript and I remarked that was withdrawn by you at HESS (Hydrology and Earth System Sciences). After that I excluded your manuscript submitted HESS and bibliography and you still have a similarity index of about 30%. I think this is a quiet big issue from my point of view. I consider you should rewrite some paragraphs in order to avoid plagiarism.
Why I consider this way?
Because you have enough data to extract the best conclusions and to strat some good quality discussions.
Ok. Maybe your research is not high in terms of novelty, but you are completing with information a white spot, let s say.
My recomandation is to split RESULTS from DISCUSSION and CONCLUSIONS.
I appreciate that the best way to bring the light and red wire in this valuable manuscript is to split this 3 sections.
If you rewrite the manuscript according to this recomendations, I reccomend it for publication.
I consider that a good database (as you have here) must be exploited to the maximum.
Reviewer 3 Report
In this study the Authors investigated the impact of the Tree Gorges Dam (TGD) closure on the mid channel bars characteristics (such as the number, area, shape, and temporal dynamics) in the middle and downstream reaches of the Yangtze River. The analysis was based on Landsat images obtained between 1985 and 2018. Image analysis was supplied with river and sediment discharge data from gaging stations located along the river. The integration of this type of data is essential for this type of analysis. The authors described in detail the initial process of image selection, preprocessing as well as the further spatiotemporal statistical analysis of different channel bar features. The Authors discussed the accuracy of extraction of channel bars and limitations of the image analysis, and suggested possible future improvements.
The Authors aptly noted and took into account area changes of the mid channel bars that might result from the changes in water level. They also discussed different factors, other than the TGD operations (such as vegetation growth, sand mining, agriculture, urbanization), that may be responsible for temporal dynamics of channel bars.
The results showed that the number of mid channel bars showed an increasing trend before the TGD operation and decreasing after the TGD closure. It should be noted that Authors analysed sand bars characteristics depending on their size (i.e., they divide sand bars in 4 groups) which is a very valuable addition. The Authors noticed that the highest change in number of channel bars was observed among small size bars in the lower reach. On the contrary, they area of all bars showed increasing trend after the TGD closure. The results showed also that large size bars seem to be less affected by the TGD operations than small bars. The Authors noticed that small size and middle size bars became more round in shape over time than large and extra-large bars that became narrow and elongated in shape. The shape of the large size bars was more stable than those of small size. The Authors showed that the impact of the TGD on mid channel bar dynamics was diminishing with the distance from the TGD.
The paper is well organized, results and methods are clearly presented and consistent. However, the text require minor English language corrections to make it more understandable to the reader. The overall contribution of the paper is sufficient for the journal and it can be published after minor revision.
Minor issues:
Line 20: ‘hydrological processing’ sounds a bit odd. Would be better to say ‘hydrological processes’.
Lines 42-43: Please use ‘channel forms’ instead of ‘features’
Line 62: It should read ‘practices’
Lines 62-63: It should read ‘which experience’
Line 85: It should read ‘river regulation’
Line 98: Please, change into ‘spatiotemporal scale’
Line 121: It should read ‘in the world’
Line 127: Could you please specify the average width of the river and its average slope in the analysed river reach?
Line 167: Please, remove ‘are’
Line 169: It should read ‘presence’
Line 180: It should read ‘the presence of MCBs’
Line 189: It should read ‘river and suspended sediment discharges’
Line 192: It should read ‘daily river discharge’
Line 194-196 Please verify the sentence. Taking into account the depth of the river I suppose that velocities were measured in the whole cross-sections using Acoustic Doppler Current Profiler (ADCP). While SSC was probably estimated from the acoustic backscatter and sediment sampling. If so, please either verify this detailed description or remove it.
Line 245-247: Please, remove ‘after carefully inspected by the first author’
Line 282: It should read ‘that occurred in the HSR images’
Line 290: It should read ‘which could result from some’
Line 306-307: Please, remove ‘that’
Line 308: Please, change as follows ‘in the area (size) and shape (LWR)’
Line 309: Please, change as follows ‘area and shape’
Line 310: Please, expand the shortcut in bracket ‘(Root Mean Square Error)’
Line 315: Would be better to say either ‘general overview of MCBs’ or ‘general picture of MCBs’.
Line 316: Please, remove ‘us’
Line 341: It should read ‘spatiotemporal’
Line 352: Please, change into ‘relatively denser than in’
Line 358: Please, use ‘occur’ instead of ‘present’
Line 380-381: Please, replace the ending as follows ‘completely disappeared or appeared.’
Lines 403-405: Please, rewrite the sentence. It is not clear.
Line 453: It should read ‘less variable and more stable’
Line 483: It should read ‘river discharge’
Line 492: It should read ‘vary with the distance’
Line 492: Please, remove ‘here’
Line 495: It should read ‘counted in ascending order’
Line 513: It should read ‘may be unreasonable’
Line 518: It should read ‘Sand Mining Management’
Line 520: Please, remove ‘happened’
Line 528: Please, remove ‘change’
Line 530: It should read ‘out of’
Line 560: Please, change into ‘statistical point of view’
Line 578-589: It seems redundant. Should be removed.
Round 2
Reviewer 2 Report
I appreciate that this manuscript was improved.
I recomend it for publication.
All the best